# Anti-Müllerian Hormone Concentrations for Determining Resumption of Sertoli Cell Function following Removal of a 4.7 mg Deslorelin Implant in Tomcats

**DOI:** 10.3390/ani13162552

**Published:** 2023-08-08

**Authors:** Lluis Ferré-Dolcet, Matteo Bordogna, Barbara Contiero, Christelle Fontaine, Silvia Bedin, Stefano Romagnoli

**Affiliations:** 1Department of Animal Medicine, Production and Health, University of Padova, 35122 Padova, Italysilvia.bedin@unipd.it (S.B.);; 2Altinia Veterinary Clinic, 30173 Venice, Italy; matteo.bordogna@studenti.unipd.it; 3Virbac, 13ème Rue, 06515 Carros, France; christelle.speiser-fontaine@virbac.com

**Keywords:** deslorelin, anti-Müllerian hormone, testosterone, resumption of fertility, tomcat

## Abstract

**Simple Summary:**

The anti-Müllerian hormone is considered the gold standard biomarker for Sertoli cell function evaluation. In male fetuses, Sertoli cells secrete high concentrations of AMH in order to suppress the development of the Müllerian ducts. This secretion is significatively reduced after puberty by the action of the follicle-stimulating hormone. Agonists of the gonadotropin-releasing hormone (GnRH) bind to GnRH receptors in the anterior hypophysis, causing an initial stimulation of luteinizing hormone (LH) and follicle-stimulating hormone (FSH) release until a desensibilization of the receptors occurs. In tomcats, testosterone concentrations tend to be basal (below 0.1 ng/mL) 20 days after the application of a 4.7 mg deslorelin implant with an average length of 18 ± 3 months. All the effects caused by the effect of the deslorelin molecule are completely reversible after the implant activity or after its removal. The present study shows that AMH concentrations increased from 20.95 ± 4.97 ng/mL to 82.41 ± 14.59 ng/mL and decreased to 28.42 ± 7.98 ng/mL three weeks after removal of a 4.7 mg deslorelin implant. Achieving pre-treatment AMH concentrations 3 weeks after removal of the implant can be considered Sertoli cell activity resumption.

**Abstract:**

**Background:** Deslorelin implant use in cats is a medical alternative to surgical sterilization, and due to its prolonged efficacy, its use has shown growing interest in the veterinary community. In the case of breeding facilities, its removal is often requested for the early restoration of testicular function. As anti-Müllerian hormones (AMH) in males is dependent of testosterone secretion, its assay may determine the restoration of testicular steroid secretion. An average of 3 weeks has been already described for tomcats’ testicular function resumption after implant removal, but information about AMH concentrations in deslorelin-treated tomcats is lacking. **Methods:** Fourteen tomcats were treated for temporary suppression of fertility with a 4.7 mg deslorelin implant, which was surgically removed after 3, 6 or 9 months (n = 6, 4 and 4 tomcats, respectively). A general clinical and reproductive check with a gonadorelin stimulation test for testosterone determination was performed before deslorelin implant administration. After implant removal, tomcats’ testicles were ultrasonographically checked for volume determination every 1-2 weeks with observation of the glans penis (presence or absence of spikes) and blood collection to assay both testosterone and AMH concentrations. **Results:** AMH concentrations increased significantly during the deslorelin treatment from 20.95 ± 4.97 ng/mL to 82.41 ± 14.59 ng/mL (*p* < 0.05). Following implant removal, AMH concentrations progressively decreased to pre-treatment levels, with a value of 28.42 ± 7.98 ng/mL on the third week post-removal where testosterone secretion was again detected. **Conclusions:** Even if a big variability of AMH concentrations exists between male individuals, resumption of tomcats’ testicular function following a deslorelin treatment can be determined by AMH assay.

## 1. Introduction

Anti-Müllerian hormone (AMH) or Müllerian inhibiting factor is a glycoprotein belonging to the group of the transforming growth factor-β produced exclusively by the gonadic somatic cells [1]. AMH is considered a Sertoli, Leydig and granulosa cell function biomarker that allows one to detect the presence of gonadal tissue in both male and female dogs and cats [2,3,4,5,6]. During fetal life, the immature Sertoli cells secrete high concentrations of AMH in order to suppress the development of the Müllerian or paramesonephric ducts and the future female reproductive tract, therefore allowing the development of the male genital tract [2,7]. In early postnatal males, AMH secretion is independent of gonadotrophic hormone production and then slowly starts to be stimulated by follicle-stimulating hormone (FSH) [8,9,10] until puberty, at which time AMH decreases as a result of Sertoli cell maturation and testosterone production exerting a negative feedback on pituitary FSH release through the androgen receptor [11,12,13]. The negative feedback of testosterone on the hypothalamic–hypophyseal–gonadic axis and on AMH production by the Sertoli cells has been demonstrated in deslorelin-treated dogs, in which testosterone concentrations decrease while AMH concentrations increase [9,14,15]. Little is known about AMH in companion animals, especially in males, where variable serum concentrations can be found between individuals and during different ages. In fact, in male companion animals, AMH has been used for cryptorchidism confirmation, absence of gonads, confirmation of testicular sertolioma, testicular atrophy and disorders of sex development [3,13,15,16,17]. In male individuals, there is a large variation of AMH concentrations, which is a challenge in determining a relationship with fertility. Indeed, studies in infertile dogs and cats and their AMH concentrations are lacking.

In the last years, increasing interest in feline breeding has been observed around the world. Some problematics might occur in tomcats housed in cat-breeding centers, where the presence of more than one tomcat might cause inter-male aggression or where, during breeding season when queens present estrus signs, the weight loss of males can be a health concern. Deslorelin implants have been approved for use in male cats and provide them with the possibility of neutering reversibility at the end of its effect or even if a prompt removal of the implant is made for a sooner restoration of testicular function.

Deslorelin is a synthetic agonists of gonadotropin-releasing hormone (GnRH), which has been widely used to cause reversible sterility in dogs, cats and ferrets for the medical control of reproduction [18,19,20,21,22,23,24,25,26,27,28]. The deslorelin molecule has seven times more affinity to the endogenous GnRH receptors of the pituitary and has a potency that is 100 times higher than endogenous GnRH [29]. The effect of deslorelin implants is considered biphasic. At the beginning, GnRH agonists binding to the GnRH receptors of the pituitary cause a strong release of both FSH and LH, causing a further increase in testosterone secretion that lasts approximately a maximum of two weeks [30,31,32]. This effect has been called the “flare-up effect”. After the stimulation process, testosterone secretion rapidly decreases and can be found to be basal (below 0.1 ng/mL) for several months. The time between implantation and the elimination of testosterone secretion is called “time to downregulation” and can last a minimum of 10 weeks to a maximum of 16 weeks in tomcats [22,25,33]. Its effects on the hypothalamic–pituitary–gonadal axis of tomcats and its complete reversibility have been well described in numerous studies [25]. When administered a 4.7 mg deslorelin implant, cats undergo a period of reproduction control (no mounting, urine marking, spermatogenesis or roaming behavior) [25,27,31,33,34,35] whose duration varies between 6 and 21 months [25,26,27,28]. As this effect could be very long for a cat breeder, removal of the implant for an early restoration of testicular function could be required. In these cases, deslorelin implants are preferably placed near the umbilical scar, in order to be easily located with palpation, facilitating surgical removal. Together with the elimination of testosterone secretion, the testicular volume decreases up to 85% of the pre-treatment volume [35]. In fact, the decrease of testicular volume can be used as an indicator of treatment effectivity, even if testosterone serum concentrations are considered the gold standard for determining testicular activity. In deslorelin-treated queens, variations in the AMH concentrations have been observed during treatment and after removal (comparable to the end of the implant effect) [36], but, to date, no information is currently available on AMH secretion in deslorelin-treated tomcats and in the restoration of testicular activity.

Thus, the aim of the present study was to evaluate the relationship of testosterone secretion and AMH before, during and after the removal of a subcutaneous slow-release 4.7 mg deslorelin implant in adult tomcats to determine the concentration of AMH at which testicular parenchyma resumes spermatogenesis.

## 2. Material and Methods

The current study was approved by the University of Padua Ethics Committee (Project n. 323548). Frozen stored samples at −20 °C from fourteen intact, privately owned adult tomcats were used. The samples belong to another study where the restoration of testosterone secretion after the use of a deslorelin implant was studied [35]. The study was conducted in the Veterinary Teaching Hospital of the University of Padua, Italy. Requirements for the tomcat to be included in the study were: to be sexually mature showing androgen behavior and secondary sexual characteristics, to be presented for temporary suppression of fertility and to have negative tests to feline immunodeficiency and feline leukemia virus.

Briefly, on the study performed by Ferré-Dolcet et al., (2020), 4.7 mg deslorelin implants (Suprelorin^®^, Virbac, Carros, France) were placed subcutaneously 1.5 cm cranial to the umbilical scar for easier localization and removal, which was performed surgically by a 2 cm incision at 3, 6 or 9 months (randomly divided) following IM treatment with dexmedetomidine (0.008 mg/kg, Dexdomtor, Orion Pharma S.R.L., Milan, Italy), ketamine (2 mg/kg, Imalgene 1000™; Merial, Padua, Italy) and butorphanol (0.3 mg/kg, Dolorex™; MSD, Rome, Italy) with IV Propofol (Proposure™; Merial, Padua, Italy) if required. Surgical incisions were closed with an intradermic suture with absorbable material (3/0 Monosyn™; Braun, Milan, Italy). Following implant removal, tomcats were checked weekly, looking for penile spikes at their glans penis, measuring variations in testicular volume by the Hansen’s formula (Volume = Length × Height × Width × 0.71) [37,38] and performing a GnRH stimulation test and collecting blood samples for hormonal assays (both testosterone and AMH).

### 2.1. Serum Testosterone and AMH Determination

Two milliliters of blood samples were obtained from the jugular vein 60 min after a GnRH stimulation test with 50 μg of gonadorelin (Fertagyl™; Intervet, Milan, Italy) administered IM in the rear limb for a serum testosterone and AMH assay. Blood samples were collected in glass tubes (BD Vacutainer^®^; BD-Plymouth, Plymouth Pl6 7BP, UK) and allowed to clot for five minutes at room temperature. Samples were centrifuged at 2500× *g* for 10 min, and serum was used for the hormonal assay. Hormonal concentrations were measured the day of implantation, the day of implant removal before surgical procedure (at 3, 6 or 9 months after implantation) and every 7 days after implant removal until testosterone secretion was restored (also evaluated by the presence of penile spikes). Serum testosterone concentration was determined by chemiluminescence (coefficient of variation: 16%) (Immulite 1000; Siemens, Milan, Italy), while serum concentration of AMH was determined with an ELISA test (AMH Gen II Elisa™, Immunotech s.r.o., Prague, Czech Republic).

### 2.2. Statistical Analysis

AMH values for treated tomcats were evaluated with a statistical package (SAS 9.4, SAS Institute Inc., Cary, NC, USA) and analyzed using a variance model with random and repeated animal effects. The treatment, time of the treatment, age of the tomcat and the interactions of time by class of age and of treatment by class of age were included in the analysis. Time was indicated as “time 0” (day of implantation), “1” (implant removal), “2” (day 8 post-removal), “3” (day 15 post-removal), “4” (day 22 post-removal) and “5” (day 29 post-removal). Animals were divided into younger or older than 12 months of age when enrolled into the study.

## 3. Results

A previous study [35] showed that deslorelin implants are extremely fragile and that accurate care during removal should be taken to avoid their rupture and to leave a small piece of the implant that might still act as treatment. Testosterone secretion in these cats was basal (testosterone < 0.1 ng/mL) at the time of implantation and was restored in all tomcats at around 3 weeks post-implant removal, as confirmed by the appearance of penile spikes, with no differences between treatment durations in terms of the time to the restoration of secretion of testosterone and the appearance of penile spikes. The body weight of every cat during the treatment increased by an average of 0.11 ± 0.04 kg (0.95%), 1.93 ± 0.27 kg (33.5%), and 1.66 ± 0.77 kg (47.8%), depending on the treatment length (3, 6 or 9 months, respectively). Testicular volume was reduced by 46.61 ± 27.92% (*p* > 0.05) without differences between treatment groups. Twelve of the fourteen treated cats were surgically neutered once testicular function restoration was detected after a GnRH stimulation test. Orchiectomy was performed within the range of the 14 days following the testosterone concentrations detection. Testicles were fixed in 4% paraformaldehyde for 48 h at room temperature and then embedded in paraffin blocks. Paraffined sections that were 4 μm thick, obtained with a rotary microtome (RM2145, Leica, S.P.A., Milan, Italy), were stained with hematoxylin-eosin and evaluated histologically with a light microscope (40×), showing different degrees of spermatogenesis depending on the interval between detection of the first testosterone secretion and orchiectomy (2 to 14 days after testosterone secretion detection). More advanced spermatogenesis phases were observed in tomcats that were neutered later after restoration of testicular function, observing all cell lines of the spermatic cycle (presence of spermatogonia, primary spermatocyte, secondary spermatocyte, spermatids and spermatozoa) in the seminiferous tubule lumen. Tomcats neutered during the first week after testosterone detection only showed the first phases of spermatogenesis (spermatogonia, primary spermatocyte, secondary spermatocyte) in the seminiferous tubule.

### AMH Concentrations

The average level of AMH on the day of implant insertion (Time 0) was 20.95 ± 4.97 ng/mL, while on the day of implant removal (Time 1) it peaked up to 82.41 ± 14.59 ng/mL (*p* < 0.05). During follow-up, mean AMH concentrations decreased to 62.57 ± 21.35 ng/mL (*p* > 0.05), 40.27 ± 7.67 ng/mL (*p* < 0.05), 28.42 ± 7.98 ng/mL (*p* < 0.05) and 30.18 ± 8.37 ng/mL (*p* < 0.05) at 1, 2, 3 and 4 weeks post-removal (Time 2, 3, 4 and 5, respectively) (Figure 1). At 3 weeks post-removal, independently of the length of the implant treatment, penile spikes and testosterone secretion resumption were detected, which was considered a criteria for study conclusion. AMH concentrations were found to be higher in tomcats younger than 12 months in comparison with tomcats over 12 months of age at the time of implantation (*p* < 0.05) (Figure 2).

## 4. Discussion

To the authors’ knowledge, this is the first study documenting AMH concentrations in tomcats during a deslorelin treatment and following removal of the implant to monitor resumption of endocrinological testicular function and spermatogenesis together with testosterone secretion resumption. In adult intact tomcats, AMH concentrations were shown to range between 4.8 and 81.3 ng/mL, while in neutered tomcats, AMH concentrations were basal, as they were found to be below the lowest standard point of the test (<0.14 ng/mL) [3]. In male testes, AMH secretion from the immature Sertoli cells peaks during fetal life, causing the regression of Müllerian ducts: in human fetuses, this process lasts from week 8 to the end of week 9, after which Müllerian ducts become insensitive to AMH [2,7]. The regression of Müllerian ducts in the male cat fetus has been observed to start when it reaches a crown-rump length of 3.2 cm [39], which, on ultrasound examination, should correspond to a gestational age of 37 days (±2 days) [40]. Postnatally, although AMH is no longer active on Müllerian ducts, its expression by Sertoli cells remains strong until puberty, except for a transient decline in the perinatal period. At the onset of puberty, AMH secretion decreases simultaneously to the onset of negative feedback provided by Leydig cells’ androgen production [3,7]. Higher AMH values in younger cats may reflect a lower or more recent exposure to negative feedback by testosterone and other androgens such as di-hydrotestosterone, while in older cats the feedback mechanism is more successful in inhibiting AMH production.

There was a great variability in AMH concentrations in the studied population (Figure 3). A similar variability was reported by Axnér and Ström Holst (2015), who reported AMH values measured in adult intact tomcats ranging between 4.8 and 81.3 ng/mL, with higher AMH concentrations in cats under 12 months of age due to the immature Sertoli cells on the testicular parenchyma [3]. Serial samples of each individual would be a good strategy in order to study inter-individual AMH concentrations that could be related to age, breed or even testosterone concentrations, as testosterone secretion is pulsatile. A study limitation might be the low numerosity of tomcats; thus, further studies on AMH concentration in cats should be performed while putting special emphasis on cats’ breeds, as this could be a determining factor. Overall, the resumption of testicular activity is not as predictable as ovarian resumption in queens who showed concentrations of 3.9 ± 0.7 ng/mL at the time of resumption of ovarian activity (which did not differ from pre-treatment concentrations) [36]. As testosterone secretion in tomcats is not related to variations of the photoperiod [41], variations of AMH concentrations cannot be attributed to the season of deslorelin implantation/removal, as is the case in female cats. To date, AMH in male companion animals has been used for the confirmation of a cryptorchidism male, evaluation of a neutered or intact status, diagnosis of testicular sertolioma, testicular atrophy and disorders of sex development [3,13,15,16,17], but the fertility potential is still under study.

With regard to the GnRH stimulation test with gonadorelin being a potential bias in our results, AMH secretion is not influenced by the GnRH stimulation test as previously observed [3]. In the present study, AMH concentrations of 28.42 ± 7.98 ng/mL indicated a complete resumption of testicular activity in tomcats older than 12 months (as in all the tomcats, even if implanted at an early age, removal was performed after they were 12 months old due to the length of the treatment of the study).

To date, only one study showed variations of AMH concentrations after the suppression of the pituitary–gonadal axis in male dogs [14] and showed an increase in its concentration starting from the moment when azoospermia was detected. Unfortunately, in the present study, AMH concentrations in the deslorelin-treated cats were only measured before the treatment, the day of the implant removal and after the implant removal. With this in mind, further studies might be performed in order to detect variations in AMH concentrations in tomcats during a deslorelin treatment in order to observe the suppression of Sertoli cell activity after treatment. In addition, further studies might be needed in which AMH concentrations are studied after the removal of a deslorelin implant until the spermatogenic cycle is complete. These variations might show AMH concentration variations dependent on the increase in testosterone secretion across time after the treatment effect.

## 5. Conclusions

In conclusion, in the present study, the deslorelin-induced suppression of testosterone production by the Leydig cells removed the negative feedback on AMH secretion, whose concentration rose significantly during treatment, reaching levels corresponding to fetal life. Once implants were removed, serum AMH concentrations started a progressive decrease to pre-treatment values, which was complete at 3 weeks post-implant removal; since testosterone production begins at the end of the “critical phase” (initiation of embryologic testis development), the AMH values of the present study might be considered equivalent to those that would be present in male cats during embryogenesis. Nevertheless, further studies are needed to evaluate the great variability in AMH concentrations between individuals.

## Figures and Tables

**Figure 1 animals-13-02552-f001:**
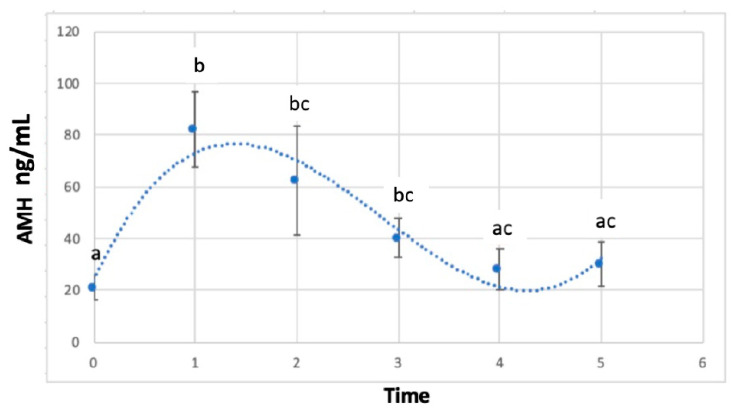
Average AMH concentrations (ng/mL) in fourteen tomcats treated with a 4.7 mg deslorelin implant which was removed after a period of 3-, 6- or 9-months subjects at time 0 (implantation), 1 (removal) and 2, 3, 4, 5 (weekly serum sample following implant removal). Different letters show the significant difference (*p* < 0.05).

**Figure 2 animals-13-02552-f002:**
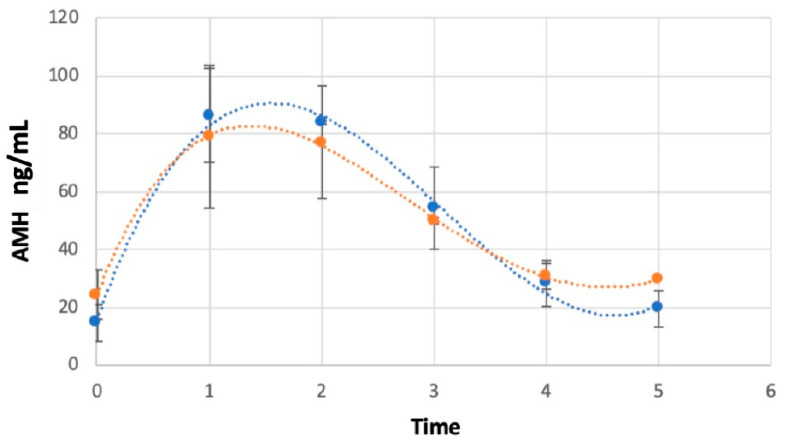
Average AMH concentration (ng/mL) as a function of time 0 (administration of a 4.7 mg deslorelin implant), time 1 (implant removal), times 2, 3, 4, 5 (weekly serum sample following implant removal) in younger and older cats. The blue line represents the curve of the tomcats under 12 months of age (n = 8), while the orange one represents the tomcats over 12 months (n = 6).

**Figure 3 animals-13-02552-f003:**
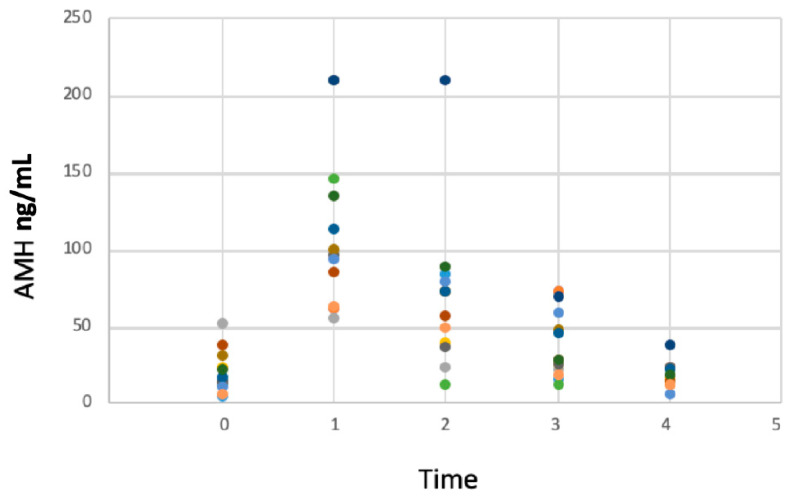
Individual AMH concentration (ng/mL) as a function of time 0 (administration of a 4.7 mg deslorelin implant), time 1 (implant removal), times 2, 3, 4, 5 (weekly serum sample following implant removal) of 14 tomcats.

## Data Availability

Data might be available under request.

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
