# Peer review of "Anti-Müllerian Hormone Concentrations for Determining Resumption of Sertoli Cell Function following Removal of a 4.7 mg Deslorelin Implant in Tomcats"

_animals, 2023, doi:10.3390/ani13162552_

Round 1
Reviewer 1 Report
The present manuscript, entitled “Anti-Müllerian Hormone concentrations for determining resumption of Sertoli cell function following removal of a 4.7 mg 3 deslorelin implant in Tomcats.”, was thoroughly revised. The topic is of great scientific and clinical interest, the paper has the right scientific soundness. Some minor revisions were made, together with some comments. English should be revised.
Line 4: Tomcats in the title with a lowercase - delete the full stop at the end of the title
Line 14: the word “SUMMARY” is not in bold
Simple summary should be expanded according to the manucript.
Line 24: “Background” should be in bold, “Deslorelin” should not be in bold. “Implat”: change with “implant”
“in cats is growing interest in due its prolonged efficacy”: please rephrase
Line 26: the title cites “Anti-Müllerian hormone” (also in keywords) the in this part is named as “Anti Mullerian Hormone”. It should be cited in the same way throughout the entire manuscript. I suggest to use the form of the title
Line 29: “Tomcat’s” change with “tomcats” - add “were” before “treated”
Line 30: add a space between “4.7” and “mg”
Line 33: tomcats’ testicles
Line 37: remove a space before “Following”. After “removal” add a comma
Line 38: pregresively change with “progressively”
Line 40: tomcats’
Line 39-41: please rephrase
Line 47: delete a space after “glycoprotein”
Line 49: delete “it”
Line 50: change with “male and female”
Line 53: add a comma after “tract”
Line 66-67: please rephrase with a more appropriate english to be clearer for the reader
Line 67: use “indeed” instead of “in fact”
Line 76: change with “a sooner restoration of testicular function”
Line 96: “In those cases, deslorelin implants should be placed near to the umbilical scar in order to have a better digital localization and surgical removal.” Change with “In this cases, deslorelin implants are preferably placed near the umbilical scar, in order to be easily located with palpation and facilitating the surgical removal”
Line 100: change “on” with “of”
Line 109: use “tomcats” throughout the entire manuscript. Change properly troughout the entire text
Line 115-117: please rephrase
Line 125: delete the “-“ between “9” and “months”
Line 127: change “Padova” with “Padua” where necessary
Line 136: remove “:” from subtitles everywhere in the text
Line 161: change with “A previous study”
Line 171-173: please rephrase deleting repetitions
Line 175: add a comma after “Italy)”
Line 188: add a comma after “post-removal”
Subchapter “AMH concentrations”: I cannot see Figures cited by Authors, and from the platform I do not see additional files. I think there were some problems with the upload, however they should be added.
Line 196: change with “authors’ knowledge”
Line 201: change with “In male testes”
Line 202: change “foetal” with “fetal” throughout the entire manuscript (also with alike terms)
Line 205: change with “when it reaches”
Line 209: “peri-natal” change with “perinatal”
Line 216: add a comma after “(2015)”. Then change “whose” with “who reported”
Line 217: “ranged” change with “ranging”
Line 225: if there is a reference to add, it would be better to change “happens” with “reported” and then add the reference at the end.
Line 227-230: please, rephrase and add some commas
Line 233: change with “as previously observed” and add the reference between squared brackets. Also change “our study” with “the present study”
Line 237-239: please rephrase. I would also suggest to the Authors to implement this part, like this it seems broken off, whilst the comparison with dogs is worth the discussion.
Line 240: change with “CONCLUSIONS”
Line 241: change “our study” with “the present study” - do it throughout the entire manuscript
Line 243: add a comma after “treatment”
Line 244: add a comma after “removed”
Line 246-248: I suggest to the Authors to explain a little bit better this part (maybe extending it) and move it to Discussion
Some comments:
Explain better the part about the inter-individual variability - is it common to find such variability? Was it reported also for other hormones? (the answer is yes :-) )
Add a comment in “Discussion” about the scarcity of cats enrolled (a flaw of the study).
Add a comment in “Discussion” about the potential use of this hormone in the clinical activity - as briefly mentioned in the introduction.
Line 284: 2005 in bold
Line 313-315: pages are missing
Line 323: year in bold
Line 328: pages are missing
Line 333: year in bold
Only minor revisions suggested.
Author Response
Please find attached the response

Reviewer 2 Report
The paper is interesting. It is based on a previous work of the author. And it was completed by adding the dosage of the hormone AMH. The materials and methods and results are pretty confusing. The reader cannot see the data of individual cats. He cannot see the variability mentioned in the text; It is not possible to understand exactly which statistical test was carried out and which error was used. Histological examinations are also described but it is not clear exactly how much time passes between the removal of the implant and the castration These histological examinations are not adequately described and are not presented as images. It is not clear how the hormone AMH can be used to evaluate the resumption of testicular function. The data can be better organized and better presented. In any case, these are Interesting results, absolutely new for the species and after treatment with deslorelin acetate.
There are many errors, We recommend a deep review of the text.
Author Response
Please find attached the response

Round 2
Reviewer 2 Report
Sorry, but I cannot see Figure 2, even in the supplementary files. In the meantime, the symbols (+/-) are not visible in the abstract. Simple abstract seems without results and discussion. I think that two lines are necessary.
Author Response
Dear reviewer 2, We feel really sorry that you could not saw figure 2 in the supplementary file. We have just uploaded again (tried with better quality) hoping it will not give any problem. In addition, as requested, we’ve added some comments in the simple summary. Unfortunately, in the main manuscript, symbols (+/-) are visible to us and seems also to the other reviewers… me might think is some kind of problem on the downloading of the manuscript. On the other side, we have just written them again trying to avoid the problem.
